# Diagnostic Benefit of Molecular Imaging in Patients Undergoing Heart Valve Surgery for Infective Endocarditis

**DOI:** 10.3390/microorganisms12091889

**Published:** 2024-09-13

**Authors:** Dustin Greve, Emma Sartori, Hector Rodriguez Cetina Biefer, Stefania-Teodora Sima, Dinah Von Schöning, Frieder Pfäfflin, Miriam Songa Stegemann, Volkmar Falk, Annette Moter, Judith Kikhney, Herko Grubitzsch

**Affiliations:** 1Department of Cardiothoracic and Vascular Surgery, Deutsches Herzzentrum der Charité, Augustenburger Platz 1, 13353 Berlin, Germany; 2Charité—Universitätsmedizin Berlin, Corporate Member of Freie Universität Berlin and Humboldt-Universität zu Berlin, Charitéplatz 1, 10117 Berlin, Germany; 3Department of Cardiac Surgery, University Hospital Zurich, 8091 Zurich, Switzerland; 4Department of Cardiac Surgery, City Hospital of Zurich, 8063 Zurich, Switzerland; 5Center for Translational and Experimental Cardiology, University of Zurich, 8091 Zurich, Switzerland; 6Department of Orthopaedic, Trauma and Plastic Surgery, University of Leipzig Medical Center, 04103 Leipzig, Germany; 7Department of Microbiology, Labor Berlin—Charité Vivantes GmbH, 13353 Berlin, Germany; 8Department for Infectious Diseases and Critical Care Medicine, Charité—Universitätsmedizin Berlin, Corporate Member of Freie Universität Berlin and Humboldt-Universität zu Berlin, Charitéplatz 1, 10117 Berlin, Germany; 9Partner Site Berlin, DZHK (German Centre for Cardiovascular Research), 13125 Berlin, Germany; 10Department of Health Science and Technology, Swiss Federal Institute of Technology, 8093 Zurich, Switzerland; 11Biofilmcenter, Infectious Diseases and Immunology, Institute of Microbiology, Charité—Universitätsmedizin Berlin, Corporate Member of Freie Universität Berlin and Humboldt-Universität zu Berlin, Hindenburgdamm 30, 12203 Berlin, Germanyjudith.kikhney@charite.de (J.K.); 12Moter Diagnostics, 12207 Berlin, Germany; 13MoKi Analytics GmbH, 12207 Berlin, Germany

**Keywords:** infective endocarditis, molecular diagnostic, molecular imaging, biofilms, FISH, FISHseq

## Abstract

(1) Background: The successful treatment of infective endocarditis (IE) relies on detecting causative pathogens to administer targeted antibiotic therapy. In addition to standard microbiological cultivation of pathogens from tissue obtained during heart valve surgery, the potential of molecular biological methods was evaluated. (2) Methods: A retrospective study was performed on heart valve tissue from 207 patients who underwent heart valve surgery for IE. FISHseq (fluorescence in situ hybridization combined with 16S rRNA gene PCR and sequencing) was performed in addition to conventional culture-based microbiological diagnostics. The diagnostic performance of FISHseq was compared with the conventional methods and evaluated in the clinical context. (3) Results: Overall, FISHseq provided a significantly higher rate of specific pathogen detection than conventional valve culture (68.1% vs. 33.3%, *p* < 0.001). By complementing the findings from blood culture and valve culture, FISHseq was able to provide a new microbiological diagnosis in 10% of cases, confirm the cultural findings in 24.2% of cases and provide greater diagnostic accuracy in 27.5% of cases. FISHseq could identify a pathogen in blood-culture-negative IE in 46.2% of cases, while valve culture provided only 13.5% positive results (*p* < 0.001). (4) Conclusions: This study demonstrates that using FISHseq as an additional molecular biological technique for diagnostics in IE adds substantial diagnostic value, with potential implications for the treatment of IE. It provides pathogen detection, especially in cases where conventional microbiological cultivation is negative or inconclusive.

## 1. Introduction

Infective endocarditis (IE) is a life-threatening infection of the heart valves or other structures of the cardiac endothelium. This disease is associated with high mortality and severe complications [1]. In most cases, the infection is caused by bacteria, rarely fungi. Identifying the causative microorganism in the affected heart valve is essential for directing antibiotic treatment. Ideally, the microorganisms can be identified from blood and valve cultures and subsequently be tested for susceptibility to different antibiotic agents. However, in many IE patients, conventional microbiological cultures return negative, e.g., due to previous antibiotic therapy or the fastidious nature of the causative microorganisms. The reported frequency of blood-culture-negative infective endocarditis (BCNIE) ranges from 2.5 to 31% [2]. BCNIE is associated with an increased overall mortality [3]. In addition to antibiotic therapy, surgical treatment of patients is required for acute IE in 25–50% of all cases and for 20–40% during recovery [4]. The gold standard for pathogen detection consists of obtaining blood cultures before starting antibiotic therapy and submitting the intraoperatively resected valve tissue for histopathological and microbiological evaluation. According to the guideline recommendations, the findings of these valve cultures should be used to direct the choice and duration of postoperative antibiotic treatment [1,5]. However, the sensitivity and specificity of the valve culture findings are reported to be low, and the positive detection rate is only 6–26% [6]. Pathogen detection in conventional culture-based microbiological diagnostics may be limited when the microorganisms elude detection. Consequently, antibiotic therapy cannot be targeted. Some possible reasons for this are previous antibiotic treatment, pathogens that are difficult or impossible to cultivate or the organization of bacteria into biofilms. This substantiates the desire for a more comprehensive and reliable examination of heart valve tissue. Over the years, molecular biological techniques have evolved and have been increasingly applied to support diagnostics in IE. But their availability is limited, and the current guidelines only mention their use in culture-negative cases without giving specific recommendations. The most widely used molecular biological methods are nucleic acid amplification and sequencing, either by polymerase chain reaction (PCR) with Sanger sequencing or by microbiome analyses. However, these methods cannot distinguish between living and dead bacteria and do not provide a spatial resolution of pathogens to detect biofilms. In addition, these techniques can produce results that are difficult to interpret if they detect microorganisms of the skin flora, which may originate from contamination of the probe.

FISHseq (the combination of fluorescence in situ hybridization with amplification and sequencing of the 16S rRNA gene) is a microscopic method that combines the advantages of molecular biology, pathology and microbiology. Fluorescence-labeled probes bind sequence-specifically to the ribosomes of the microorganisms, which are thus identified independently of the culture. FISH makes the microorganisms spatially visible and simultaneously characterizes them in the tissue context of the sample. Since FISHseq has been established [7], it has been frequently used in native and prosthetic valve endocarditis [8,9,10,11]. This method allows simultaneous pathogen identification and visualization in situ and has shown promising results for improvements in the diagnosis of endocarditis in past studies [7,8,10].

The present study evaluated the diagnostic potential of FISHseq in addition to conventional microbiological culture.

## 2. Methods

### 2.1. Patient and Case Definition

We retrospectively evaluated 554 consecutive patients undergoing cardiac surgery for definite or possible IE in the Department of Cardiovascular Surgery at the Charité—Universitätsmedizin Berlin between 2010 and 2020. The Ethics Committee of the Charité—Universitätsmedizin Berlin approved the study (EA2/236/20). Heart valve samples were obtained for routine microbiological examination during all surgeries. Based on the endocarditis team’s recommendations or at the surgeon’s discretion, valve samples of 207 patients (37.4%) underwent parallel molecular biological diagnostics (FISHseq). This study analyzed the results of the conventional valve culture and FISHseq in these 207 cases to evaluate the diagnostic benefit of molecular biological techniques as an additional diagnostic tool.

### 2.2. Study Design and Statistics

The study data were obtained from a case-by-case review of the patient records. We categorized the findings of the molecular biological examinations (FISHseq) based on the visualization of microorganisms and their activity. The primary outcome measure was the rate of positive specific pathogen detection by FISHseq compared to the conventional valve cultures. Only results in which one or more particular pathogens (species, genus or family) could be detected in FISHseq were considered positive for this analysis. In cases where FISHseq provided pathologic findings but these were nonspecific without pathogen species identification, the result was assessed as negative. The diagnostic rates were further examined by subgroup, categorized by using preoperative diagnosis, preoperative blood culture findings and the duration of antibiotic treatment before surgery.

To assess the clinical impact of the additional FISHseq diagnostics, the specific constellations of findings from each diagnostic method were reviewed. By comparing the exact pathogen data, outcome groups were identified where the cultural findings could be confirmed by FISHseq, where there was a new diagnosis by FISHseq and where there was a gain of accuracy in pathogen identification. Conflicting results, e.g., different pathogens being found in FISHseq and the valve culture and valve-culture-positive cases in which no pathogen could be detected via FISHseq, were classified as inconclusive.

Descriptive statistical methods were applied to the study population as well as the study results. The data are presented as absolute and relative frequencies for categorical variables and as means with standard deviation for continuous measures. Comparison between groups of different diagnostic methods was carried out using Pearson’s chi-squared test. *p*-values < 0.05 are considered significant. The statistical analysis was carried out using SPSS (IBM Corp. Released 2020. IBM SPSS Statistics for Windows, Version 29.0. Armonk, NY, USA: IBM Corp.). The figures were created using Illustrator (Adobe Systems Software Ireland Limited, Version 2021, Dublin, Ireland). The study data were collected and managed using the REDCap electronic data capture tools hosted at the Charité—Universitätsmedizin Berlin.

### 2.3. Microbiological Analysis

#### 2.3.1. Standard Culture Methods

Blood culture sets consisting of one aerobic and one anaerobic bottle were inoculated with approximately 10 mL of blood. The blood cultures were incubated for 5 days in a continuously monitored blood culture system (BD BACTEC FX, Beckton Dickinson (Franklin Lakes, NJ, USA) and BACT/ALERT, bioMérieux (Marcy-l’Étoile, France)). When CO_2_ production was detected by the instrument, Gram staining and subculture on solid media were performed. The solid media included Columbia agar with 5% sheep blood and chocolate agar, incubated under aerobic conditions with 5% CO_2_, and Schaedler agar, incubated under anaerobic conditions (all culture media from Beckton Dickinson). Microorganisms were identified using routine microbiological techniques, including MALDI-TOF mass spectrometry and biochemical systems (Vitek MS and Vitek 2, bioMérieux).

For heart valve culture, the microbiological procedures included Gram staining and conventional culture of the tissue samples. Culture on solid media was performed as described above, with incubation of the Schaedler agar for 14 days. In addition, thioglycolate broth was inoculated and also incubated for 14 days. When growth was detected, microorganisms were identified as described above.

#### 2.3.2. FISHseq

The resected heart valve was divided into equal parts according to its macroscopic aspect in the operating room. For FISHseq, part of the heart valve was fixated with FISHopt^®^ fixation solution (MoKi Analytics, Berlin, Germany) in the operation theater and transported to the laboratory. The patient samples were embedded into methacrylate (Technovit 8100; Kulzer, Wehrheim, Germany), and consecutive histological sections were submitted to FISH and in parallel to DNA extraction with subsequent 16S rRNA gene PCR and sequencing, as described previously [7,12,13].

Briefly, the sections were hybridized with the pan-bacterial probe EUB338 (labeled with Cy3, orange), detecting most bacteria to visualize the entire active ribosome-containing bacterial population. A nonsense probe, NON EUB338 (labeled with Cy5, infrared), was also used to exclude unspecific probe binding to degraded nucleic acids. DAPI (4′,6-diamidine-2′-phenylindole dihydrochloride, blue) was used to visualize nucleic acids in both the host cell nuclei and bacteria. Autofluorescence was controlled using the green fluorescence channel without labeling. When a positive EUB338Cy3 signal was detected, a panel of genus- or species-specific probes was applied to differentiate between the microorganisms further [7,14]. These included the typical pathogens (*Staphylococcus* sp., *S. aureus*, *Streptococcus* sp., *Enterococcus* sp., *E. faecalis* and *E. faecium*) and, depending on the morphology, rare pathogens (e.g., *Tropheryma whipplei*, *Bartonella quintana*, *Coxiella burnetii* and *Cutibacterium (Propionibacterium) acnes*). An example of the above techniques is illustrated in Figure 1 based on a *Streptococcus mitis* case in BCNIE. The formation of the bacteria in situ can also be assessed using these methods, providing information on the presence of possible biofilms. Biofilm formation is shown in Figure 2 as an example of a *Staphylococcus aureus* case.

After PCR and Sanger sequencing, the sequences were analyzed using the diagnostic-grade commercial Centroid database from the program SmartGene (Version 5.3, SmartGene, Inc., Lausanne, Switzerland).

## 3. Results

### 3.1. Patients

The clinical characteristics of the study population are presented in Table 1. The majority of the patients were male (72.0%), and their mean age was 62 years. The proportion of patients with prosthetic valve IE was 35.7%. Due to comorbidity and a critical preoperative state, the calculated perioperative risk was high, with a mean EuroSCORE II of 16.1%. Preoperatively, endocarditis was classified as definite IE in 65.0% of cases and possible IE in 35.0% of cases. The preoperative blood cultures were positive 68.1% of the time. In 93.7% of cases, antibiotic therapy was started preoperatively with a mean duration of 9.7 days before the time of surgery. Aortic and mitral valve procedures, prevailingly valve replacements, were most frequently performed. Their 30-day mortality was 13.5%.

### 3.2. Pathogen Detection in FISHseq Compared to Valve Culture

As shown in Table 2, the conventional microbiological valve cultures were positive in only one-third of patients, whereas FISHseq identified causative pathogens in more than two-thirds (*p* < 0.001). When the patients were grouped by clinical diagnosis of IE, FISHseq confirmed a diagnosis for the clinical categories of definite and possible IE at significantly higher rates. For both positive and negative blood culture findings, there was a considerably higher rate of detecting causative microorganisms by FISHseq compared to conventional valve culture. Remarkably, the valve culture detected pathogens in BCNIE only in 13.5% of cases, while for FISHseq, this rate was 46.2% (*p* < 0.001). The findings in the subgroup for inconclusive or missing blood culture findings showed the same trend in positive diagnostic rates, but this was not statistically significant due to the small number of cases. The valve culture and FISHseq detection rates were also compared between native and prosthetic valve IE. Valve cultures were positive in just over 30.0% of patients with both native and prosthetic valves. FISHseq was positive in 70.7% of the samples from native valves and 66.2% of the probes from prosthetic valves. Lastly, regarding preoperative antibiotic treatment, significantly higher detection rates were found with FISHseq for both a duration of ≤5 days or >5 days of treatment before surgery. Overall, the impact of previous antibiotic treatment on FISHseq was smaller than that on conventional valve culture. The duration of preoperative antibiotic therapy negatively impacted how often traditional valve culture detected a pathogen, with only 26.7% detection in cases with pretreatment of more than 5 days. In this subgroup, FISHseq could still detect specific pathogens more than twice as often as valve culture (63.8%, *p* < 0.001).

### 3.3. Specific Pathogens Detected by FISHseq as Compared to Valve Culture

While FISHseq proved more sensitive than the valve cultures, the distribution of the pathogen groups was similar for both methods. The pathogen spectrum in the study cohort was composed mainly of streptococci, coagulase-negative staphylococci (CNS), *S. aureus* and *E. faecalis* (Figure 3). While CNS accounted for the most significant proportion of the valve culture findings, streptococci were the most common pathogens in FISHseq. In some cases, FISHseq could also detect rare IE pathogens that were not detectable in the conventional cultures. As an example, the findings of a *Corynebacterium diphtheriae* IE case are illustrated in Figure 4.

Figure 5 visualizes the relationship between the detailed findings and the pathogen constellations in the valve culture compared to FISHseq. While FISHseq confirmed the findings of the conventional valve culture in 93 patients (44.9%)—either by the detection of identical microorganism (*n* = 39) or by the exclusion of pathogens (*n* = 54)—it identified the causative microorganism in the first place in 83 cases (40.1%). Among those pathogens, 5 were fastidious organisms that are notoriously difficult to detect in conventional culture (*Bartonella* spp. *n* = 2, HACEK group *n* = 1, *Coxiella burnetii n* = 1 and *Tropheryma whipplei n* = 1). In 20 cases (9.7%), FISHseq and the valve culture provided differing results. This category included cases with contradictory results and cases where one of the methods detected multiple pathogens, which were only partially confirmed by the other method. FISHseq was negative in ten valve-culture-positive cases. In these cases, the valve culture detected CNS most frequently.

### 3.4. FISHseq and Valve Culture Results in Relation to Preoperative Blood Culture Findings

As shown above, FISHseq proved to be the most sensitive method in our cohort. Additionally, we analyzed its accuracy by comparing the FISHseq results with those of conventional culture. For the following analysis, we considered all 201 cases with FISHseq, valve culture and preoperative blood cultures available. In the majority of cases, positive findings resulted from two or more methods. All diagnostic techniques were negative in 27 cases (13.4%), and no pathogen was identified. In 25 cases (12.4%), only the preoperative blood culture but neither the valve culture nor FISHseq could detect a pathogen. In 18 cases (9.0%), FISHseq was the only method that detected a pathogen. We analyzed the results of the valve cultures and FISHseq in relation to the results of the preoperative blood cultures. When comparing the detected pathogens to the preoperative blood culture data, FISHseq and valve cultures can confirm a pathogen’s presence or absence and provide a new, different or additional pathogen. The results of the valve culture and FISHseq as they relate to the preoperative blood culture data are presented in Table 3.

While valve culture was only able to confirm the pathogen from the positive blood cultures in 28.4% of cases, FISHseq was able to do so in 66.0% of cases. As shown in Table 2, FISHseq detected a pathogen for the first time in 46.2% of the BCNIE cases, compared to valve culture doing so in 13.4% of cases. In culture-positive cases, there were 13 instances (9.2%) in which FISHseq found a pathogen different from the one detected in the blood culture. Interestingly, these pathogens were detected by PCR only but not by FISH, indicating the presence of inactive bacteria. In addition, there were two cases where FISHseq confirmed the blood culture organism and detected an additional one. Valve culture did so in one case. In cases where multiple pathogens were detected in the blood culture, the constellations of the findings were more heterogeneous. Valve culture and PCR found at least partially different organisms in all cases, while FISH did in all but two. The valve culture and FISHseq remained negative in three and one cases, respectively.

### 3.5. The Diagnostic Impact of FISHseq

To analyze the clinical impact of the FISHseq diagnostics performed in individual patients, we compared all the elements of the FISHseq results, including the activity of the bacteria in FISH and their spatial organization, with the results from the conventional cultures. From this comparison, we identified five categories of IE diagnosis (Figure 6).

In 24.2% of cases, FISHseq led to confirmation of the conventional cultural findings, i.e., in culture-positive cases (*n* = 34), FISHseq found the same pathogen, and the infection was assessed as active or (partially) treated based on microscopic DAPI- and FISH-positive images. In culture-negative cases (*n* = 16), neither FISH nor PCR found a microorganism, and the microscopic images were DAPI-negative. In 9.7% of cases, FISHseq identified a microorganism for the first time, and the infection was assessed as active or (partially) treated based on microscopic DAPI- or FISH-positive images in cases where the conventional cultures could not detect a pathogen. In 27.5% of cases, a gain in diagnostic accuracy could be achieved by FISHseq, i.e., in 50 cases, conflicting results present in the BC and VC could be clarified by FISHseq. Qualitative visual assessment of the FISH-active bacteria could distinguish between the leading causative pathogen(s) and contamination or coinfection.

Similarly, FISHseq identified a causative pathogen that was not detected by the conventional culture in three cases. In other cases, FISHseq was able to clarify contradictory results through indication of contamination of the conventional cultures. Typical skin microorganisms were detected in the conventional cultures, but FISHseq found no DAPI-positive microorganisms and a negative PCR result. In 41 cases (19.8%), the FISHseq findings allowed us to make conclusions about the course of the infection over time, specifically constellations of positive preoperative blood cultures combined with negative valve cultures and a FISH analysis that found DAPI-positive but FISH-negative bacteria. Since the FISH signal correlates directly with the ribosome content, these may be considered inactive pathogens (thus, IE is not assessed as active). This may, for example, occur after successful antibiotic treatment: in 29 of these cases (70.1%), antibiotic therapy had been administered for at least 5 days before surgery.

Lastly, there remained the category of inconclusive results for 39 cases: in 14.0% (*n* = 29) of cases, the FISHseq findings were negative despite positive findings in the conventional cultures. FISHseq found no proof of active or dead bacteria in eight cases of positive preoperative blood cultures for pathogens not commonly associated with contamination. In nine cases, FISHseq was similarly negative even though the valve cultures were positive for pathogens not widely associated with contamination. These cases may be attributed to the tissue samples needing to be obtained and preserved correctly. Additionally, 12 cases had a negative conventional culture, and FISHseq could not identify a pathogen but found evidence of inactive bacteria, suggesting a previous infection. These cases were all rated as inconclusive since the information they provided was not clinically actionable and seemed contradictory.

Overall, we found that FISH analysis can provide qualitative information regarding activity, spatial organization and microorganism formation, which can be helpful in most instances and crucial in some instances. Notably, FISH analysis can detect the presence of biofilms. Biofilms are clinically relevant as they may require different or prolonged antibiotic therapy. In our cohort, biofilms were reported in 69 (33.3%) cases. Interestingly, antibiotic therapy had already been administered in 31.0% of them for at least five days preoperatively. In eight instances, fastidious microorganisms formed biofilms.

## 4. Discussion

### 4.1. The Study Population

With 207 patients, this study provides the largest cohort to date available examining the diagnostic impact of FISHseq in IE. Overall, our results are consistent with a previous study examining a smaller patient cohort [10]. The distribution of gender and age is similar to those reported in other cohorts of IE [15]. The ratio of 25.0% of BCNIE cases is equally representative [16]. The proportion of prosthetic valve endocarditis has steadily increased in recent decades and culminated in a rate of 35.7% in this study [17]. The rate of comorbid conditions and the calculated perioperative risk (a mean EuroSCORE II of 16.1%) are high but fall within the expected range for IE. The 30-day mortality in this study cohort is 13.5%, which is lower than the expected rate of 16.1% as calculated by EuroSCORE II and lower than the 20.0% reported in a recent meta-analysis [18]. Consequently, we consider our study population a representative sample for most IE patients.

### 4.2. Identification of Pathogens in IE by FISHseq

This study demonstrated FISHseq’s ability to identify the causative microorganism in patients with IE. While intraoperatively obtained valve culture represents the gold standard, in our representative cohort, valve culture failed to detect an organism in most cases, possibly due to previous antibiotic treatment. For these valve-culture-negative patients, identifying microorganisms in FISHseq provides a significant diagnostic gain and allows us to administer a targeted antibiotic therapy. Compared to conventional culture, FISHseq achieved significantly higher rates of specific pathogen detection in our study (see Table 2). The overall diagnostic performance of conventional valve cultures in IE is well known to be relatively poor [19]. While valve cultures only identified a pathogen in 33.3% of all cases, FISHseq provided a specific positive result in 68.1% of cases (*p* < 0.001). This higher pathogen detection rate was significant in both subgroups in the analysis divided by a preoperative diagnosis of possible vs. definite IE.

At 25.0%, the proportion of BCNIE in the present study falls within the expected range. In blood-culture-negative cases, the success rate of valve culture was deficient at 13.5%, which preoperative antibiotic treatment of the patients could explain. This left 22.0% of IE cases in which the conventional methods could not detect a causative microorganism. It is therefore especially relevant that FISHseq identified a pathogen in almost half (46.2%) of the BCNIE cases. Pathogen detection via FISHseq allows targeted antibiotic therapy in BCNIE cases and thus may improve the clinical outcomes in these cases.

FISHseq’s greater sensitivity as compared to conventional cultures is also especially relevant to another subgroup of patients: those who receive antibiotic pretreatment. In IE patients, antibiotic therapy should be initiated immediately after obtaining their first blood cultures. Unfortunately, it is not uncommon in clinical practice for antibiotics to be administered before blood cultures are obtained [12]. In other instances, patients with IE are treated with an antibiotic for a different infection at the same time. A correlation between the duration of preoperative antibiotic treatment and a low probability of positive pathogen detection in conventional valve culture has been shown [20]. However, valve culture positivity indicates vital bacteria even at the time of surgery and has been demonstrated to have a strong negative impact on IE patients’ survival [21]. In cases of extensive antibiotic pretreatment, FISHseq could still identify the causative pathogen in 63.8% of cases. The disadvantage of poorer detection rates after the prompt initiation of antibiotic therapy may therefore be less pronounced in FISHseq than with conventional cultures. The ability of FISHseq to detect even avital microorganisms could help narrow down antibiotic treatment to specific pathogens where conventional microbiological culture fails.

### 4.3. Potential and Challenges of FISHseq in the Clinical Context

While conventional microbiological cultures provide information on pathogen identification and antibiotic susceptibility, FISHseq supplements these results with information about the identity of the pathogen in situ based on bacterial rRNA and DNA, the bacterial activity and amount and their formation (single bacteria, microcolonies or biofilms). The full diagnostic potential of FISHseq unfolds when interpreting its results and integrating them into those from traditional cultures.

In 24.2% of the cases in this study, FISHseq confirmed the pathogen found in the conventional culture. The distribution of the pathogen families in this study was as expected for all endocarditis cases, with streptococci, CNS and *S. aureus* being the most significant proportions. FISHseq sometimes detected rare, fastidious and atypical pathogens, which was less frequently achieved in the conventional culture. Especially for single rare pathogens like *Tropheryma whipplei*, which are not methodically detectable in conventional culture, FISHseq closes a diagnostic gap.

In 27.5% of cases, FISHseq led to a gain in diagnostic accuracy. As the method quantifies and validates its PCR results via the microscopic impression in situ using specific FISH probes, it can definitively distinguish between infection and contamination. This is relevant in cases where the blood and valve culture results show different pathogens. In these cases, FISHseq can help to identify the causative pathogen. FISHseq may also identify contamination in conventional culture when there are no typical infectious pathogens in the tissue sections of the valves. Particularly in the case of a single valve culture positive for a microorganism commonly associated with contamination, such as skin flora organisms, FISHseq can help to adjust the antibiotic treatment by excluding infection of the valve (increasing the diagnostic certainty). However, it must be noted that the distribution of the sample material in the operating room can lead to the focus of the infection being found either in the conventional culture or in FISHseq depending on the pathogen’s dispersal across the entire heart valve.

A subgroup in which the visual information provided by FISHseq is especially beneficial is in polymicrobial IE cases, which may account for up to 5.9% of all IE cases [22]. However, polymicrobial IE is not known to be associated with a worse outcome. In our cohort, there were 11 patients with multiple positive results from one of the diagnostic methods. FISHseq helps to distinguish in which of these cases contamination is likely, where actual polymicrobial IE is present and, in the case of a mixed infection, whether there is a primary pathogen responsible for the infectious activity. Excluding polymicrobial IE using FISHseq could enable further de-escalation of targeted antibiotic therapy.

In its unique combination of quantitative and qualitative methods, the direct visualization of infectious progress provided by FISHseq extends its diagnostic potential to the clinical setting. In conjunction with growth in the valve culture, conclusions about infectious activity may be drawn by distinguishing DAPI positivity (detecting avital and vital pathogens) and detectability with specific FISH probes. Information regarding the activity of endocarditis at the time of surgery could become relevant to the decision on the duration of antibiotic therapy. In our cohort, 34.8% of cases were classified as “Active IE”, i.e., there was visual proof of active bacteria in FISHseq. Depending on the duration of antibiotic pretreatment, this may indicate the need to adapt or escalate the antibiotic therapy. On the other hand, we interpreted 19.8% of cases as infectious remission, potentially indicating the effect of antibiotic pretreatment. If the valve culture is negative and the constellation of findings in FISHseq suggest treatment has been effective, earlier de-escalation or shortening of intravenous antibiotic therapy may be possible, thus facilitating therapy with fewer side effects and earlier hospital discharge of patients. However, prospective studies are necessary to address this question.

For the findings from the conventional culture and FISHseq that could not be interpreted conclusively, the therapeutic approach can only be discussed at the level of each individual case. For streptococci, it should be noted that in several cases, these differences only extended to the species or genus, not the family level. Therefore, they may have had a relatively small effect on the susceptibility to antibiotic treatment. On the other hand, the present study found ten cases in which FISHseq did not detect a pathogen while the valve culture was positive for a pathogen not commonly associated with contamination. This may be attributed to unequal dispersal of the local infection and a non-infected piece of the intraoperatively obtained material being sent in for examination. It is advisable to send in whole valve sections, including adherent vegetations, and to consider this when dividing the material for microbiology and molecular diagnostics.

There are yet to be sufficient data to determine which therapeutic consequences may arise from the information provided by FISHseq beyond identifying pathogens. While the formation of biofilms was reported in one-third of cases, it remains unclear whether antibiotics with particular biofilm activity should be considered in these cases. In the case of a native valve, the surgeon typically removes all affected parts of the valve, including possible adherent biofilms. However, treating bacteria organized in biofilms could be relevant in cases of prosthetic valve IE where possibly infected material remain in situ.

### 4.4. The Overall Role of Molecular Biological Techniques in IE

As molecular biological methods such as FISHseq become increasingly available, they are on their way to becoming essential components of a microbiological diagnosis of IE. This is reflected in the recent update (2023) of the ESC guidelines for managing endocarditis and the ISCVID Duke criteria, which now suggest using PCR and sequencing of the intraoperatively obtained material to establish a pathogen in BCNIE patients. For cases of blood-culture-negative prosthetic valve endocarditis, the guidelines additionally name FISHseq as a diagnostic tool [23]. Furthermore, the guidelines emphasize the need for more information on the accuracy of molecular diagnostics. If studies such as this can establish the diagnostic accuracy and potential of FISHseq, future recommendations may suggest an even broader use of molecular biological methods. The increased relevance of FISHseq is also reflected in the new version (2023) of the modified Duke criteria, which define FISHseq as a pathologic criterion for definite IE [24].

One major shortcoming of molecular diagnostics is their inability to provide information about microorganisms’ antibiotic susceptibility. Cultivating pathogens from blood and valve tissue is essential for targeted antibiotic treatment. In cases where only FISHseq detects a pathogen, antibiotic therapy must be based purely on expected susceptibility. As shown herein, there are occasional discrepancies between the pathogens detected in blood culture, valve culture and FISHseq. Hence, FISHseq should not be seen as a competitor with conventional microbiological cultivation but as a complementary method.

The accuracy and impact of molecular diagnostic results rely on the correct order and technique of sample taking and understandable communication of the results to the clinical team. As FISHseq’s method and its application have continuously evolved over the past decade, an integrated diagnostic course has yet to be consistently established. Here, the importance of an IE team, as emphasized in the ESC guidelines, must be stressed.

Overall, this study demonstrates the valuable additional information FISHseq provides for most IE patients. The translation of these additional findings into clinical recommendations remains challenging, not least due to the broad range of the presentation of IE cases in our study, often requiring an individual therapeutic approach. This further underlines the importance of reliable interdisciplinary collaboration between surgeons, cardiologists, infectious disease specialists and microbiologists to provide these complex patients with the best care. Particular emphasis must be placed on the FISHseq method for patients undergoing surgery for IE as an additional diagnostic option.

### 4.5. Limitations

The present study is a retrospective data evaluation. Thus, all the limitations inherent to retrospective studies apply. This is especially relevant as IE and treating patients with IE can vary significantly from case to case. Although the recommendations from the endocarditis team regarding the indications for molecular biological analysis were followed in most patients, individual decisions made by the responsible surgeon may have caused bias. As FISHseq is an assay for intraoperatively obtained valve material, it is only available for IE cases requiring surgical treatment. As FISHseq cannot provide information about antibiotic resistance, it can only be seen as an addition to conventional culture. As an infection does not necessarily affect the valve uniformly, sending separate samples for conventional valve culture and molecular biological analysis may have contributed to the divergent or inconclusive findings. Furthermore, our experience with sample taking, fixation and analysis has grown over the ten-year course of this study. This study´s IE diagnosis was based on the European Society of Cardiology’s 2015 algorithm for diagnosis of infective endocarditis [1]. The definitions within the 2023 European Society of Cardiology’s modified diagnostic criteria for infective endocarditis have yet to be applied. The possible negative predictive value of FISHseq in terms of diagnostic rule-out of IE cannot be evaluated through this study, as only confirmed IE cases were included. Although our study demonstrated the diagnostic value of FISHseq in endocarditis patients, the extent of its clinical benefit in terms of their outcomes has yet to be explored in future prospective studies.

## 5. Conclusions

In this study, using FISHseq was shown to add substantial diagnostic value in patients undergoing surgery for infective endocarditis. FISHseq provided pathogen detection in many cases where conventional microbiological cultivation of the valve specimens failed, including both native and prosthetic valves. This offers significant value, especially in cases of blood-culture-negative endocarditis and with extensive antibiotic pretreatment. Furthermore, FISHseq could often distinguish between contamination and infection in cases with contradictory results from conventional cultures. FISHseq also allowed a more in-depth assessment of the infectious activity of IE. At the individual case level, FISHseq helped clarify inconclusive results from conventional cultures. The potential of FISHseq as an additional diagnostic option in IE patients who undergo surgery needs to be emphasized and explored further and more systematically. FISHseq should be integrated into a standardized diagnostic algorithm to unfold its potential fully. Prospective studies are needed to provide specific recommendations for the use and therapeutic consequences of FISHseq diagnostics in IE.

## Figures and Tables

**Figure 1 microorganisms-12-01889-f001:**
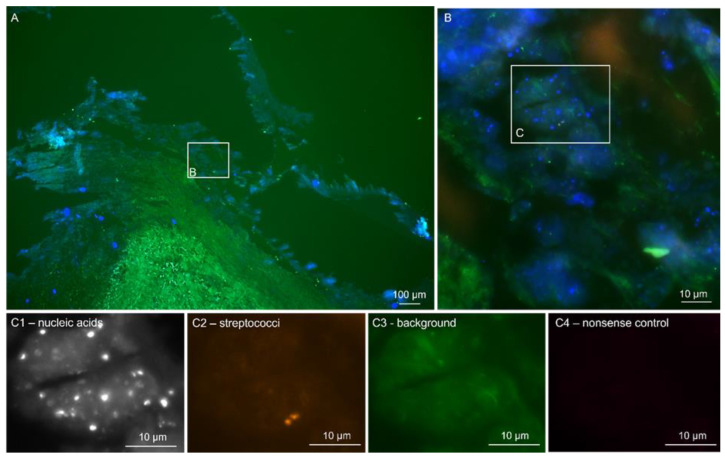
FISHseq shows active FISH-positive bacteria in blood-culture-negative endocarditis (BCNIE). Note: FISHseq analysis of a BCNIE case caused by a *Streptococcus mitis* group in an aortic valve. (**A**) Overview of the heart valve tissue (green) with host cell nuclei (nucleic acid stain DAPI in blue). (**B**) Magnification of the inset in (**A**). The autofluorescent tissue background is shown in green and the nucleic acid stain DAPI in blue. Single bacteria are positive with the streptococci-specific FISH probes STREP1/2 in orange. (**C1**–**C4**) Magnification of the inset in (**B**). The same microscopic field of view is shown as single channels. (**C1**) The nucleic acid stain DAPI in black and white shows single cocci. (**C2**) The streptococci-specific FISH probes STREP1/2 in orange show two FISH-positive active streptococci. (**C3**) The tissue background in green shows no autofluorescent structures. (**C4**) The nonsense control FISH probe NON338 shows no unspecific probe binding; 16S rRNA gene PCR and sequencing resulted in a clear sequence of *S. mitis*/*oralis* 100% over 499 base pairs.

**Figure 2 microorganisms-12-01889-f002:**
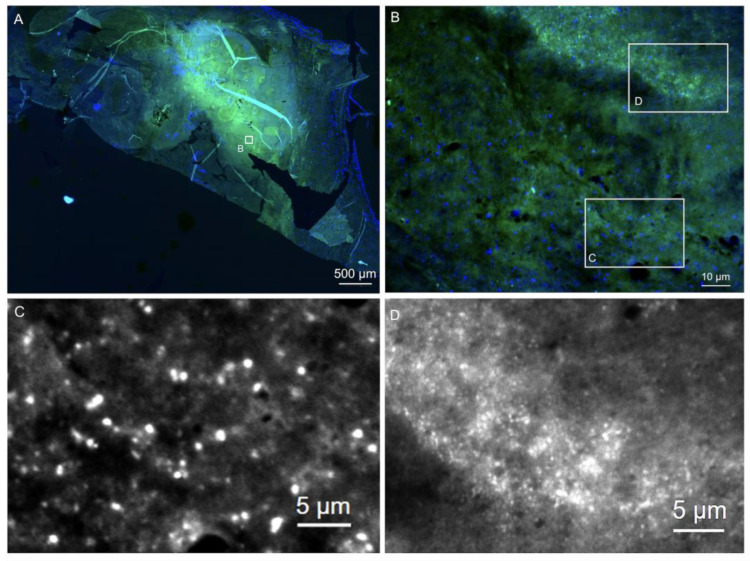
FISHseq shows bacterial biofilms in culture-negative endocarditis. Note: FISHseq revealed *Staphylococcus aureus* biofilms in a native aortic heart valve endocarditis case where both the blood and valve cultures remained negative. The labeled insets (**B**–**D**) indicate regions selected for increased magnification corresponding to respective panels. (**A**,**B**) Overview of the heart valve tissue (green) with bacterial biofilms (nucleic acid stain DAPI in blue). (**C**) The nucleic acid stain DAPI in black and white shows cocci. No FISH signal was detectable. The slightly degraded morphology of the bacteria together with the absence of a FISH signal point to antibiotic treatment of the patient before surgery. This also explains the absence of growth of the normally well-cultivable bacteria in culture. (**D**) The FITC channel shows an autofluorescent tissue background in black and white with a long exposure time to visualize the biofilm background; 16S rRNA gene PCR and sequencing resulted in a clear sequence of *S. aureus*.

**Figure 3 microorganisms-12-01889-f003:**
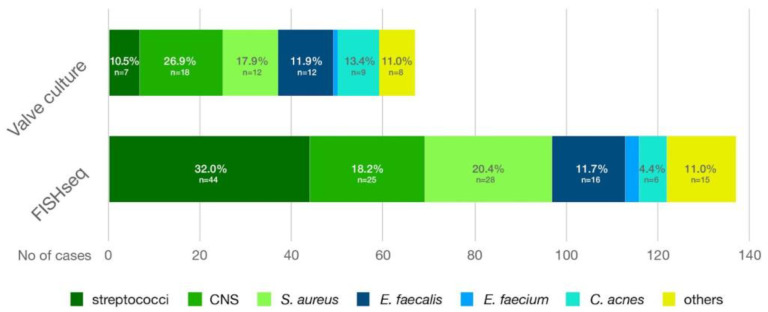
Pathogen distribution by conventional valve culture versus FISHseq. Abbreviations: CNS: coagulase-negative staphylococci; FISHseq: fluorescence in situ hybridization combined with 16S rRNA gene polymerase chain reaction and sequencing.

**Figure 4 microorganisms-12-01889-f004:**
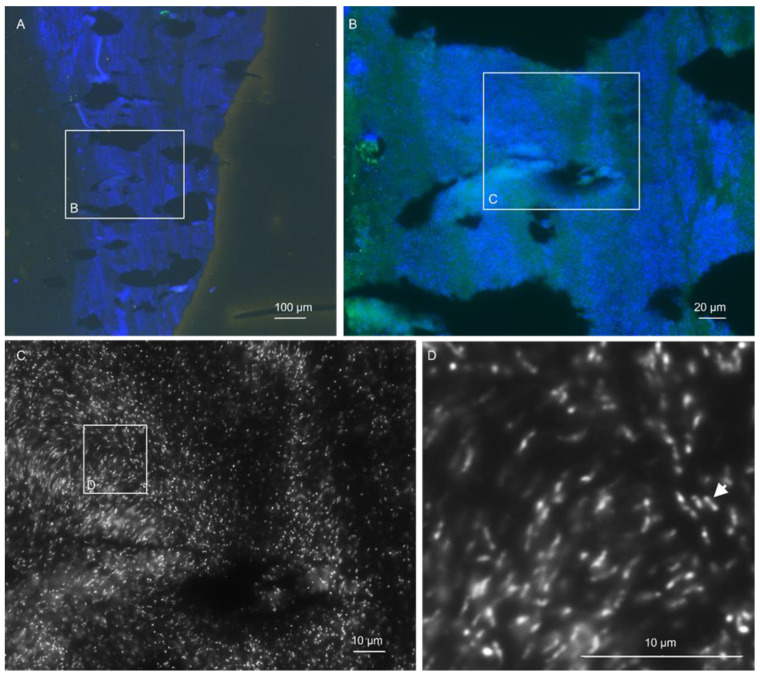
FISHseq shows Corynebacterium diphtheriae endocarditis. Note: FISHseq revealed massive biofilms of the rare endocarditis pathogen Corynebacterium diphtheriae in a case of aortic heart valve endocarditis. (**A**) Overview of the heart valve tissue with widespread bacterial biofilms (nucleic acid stain DAPI in blue). (**B**) Magnification of the inset in (**A**). The autofluorescent tissue background in green and the acid stain DAPI in blue. No FISH signal was detectable. (**C**) Magnification of the inset in B. The nucleic acid stain DAPI in black and white shows rod-shaped and pleomorphic bacteria. (**D**) Magnification of the inset in (**C**) showing single bacteria in black and white partly featuring the typical granules in the polar regions of rods (arrow); 16S rRNA gene PCR and sequencing resulted in *C. diphtheriae* (100% 459 base pairs).

**Figure 5 microorganisms-12-01889-f005:**
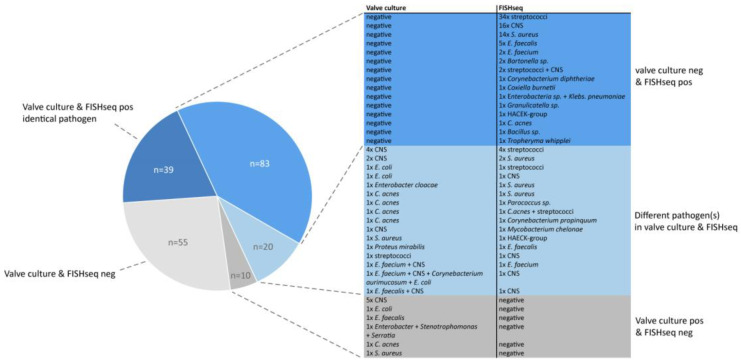
Constellations of findings from valve culture compared to FISHseq. Note: Different subspecies of streptococci and coagulase-negative staphylococci (CNS) are grouped together. Factors show how often an exact constellation occurred.

**Figure 6 microorganisms-12-01889-f006:**
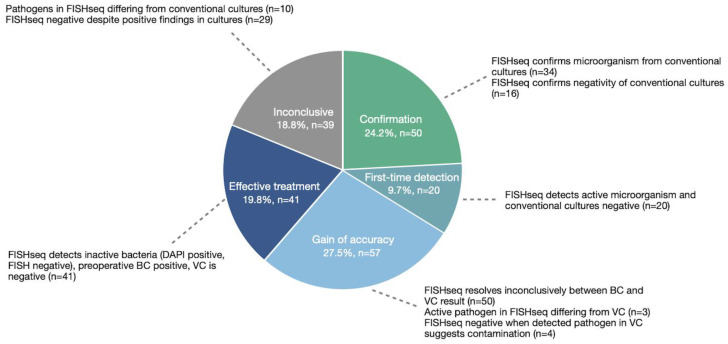
Diagnostic impact of FISHseq complementing findings from blood culture and valve culture. Abbreviations: BC: blood culture; VC: valve culture. Note: For this analysis, the results of FISHseq were related to those of both valve culture and preoperative blood culture. In the case that (for example, after successful antibiotic pretreatment) VC was negative and FISH showed DAPI-positive but inactive endocarditis, this constellation was considered “effective treatment”.

**Table 1 microorganisms-12-01889-t001:** Baseline characteristics of the study population.

*n*	207
Female (*n* (%))	58 (28)
Age (mean (sd))	62 (±15.5)
BMI (mean (sd))	25.6 (±4.6)
EuroSCORE II (mean (sd))	16.1 (±15.9)
comorbidities (*n* (%))	
Peripheral artery disease	36 (17.4)
COPD	35 (16.9)
Diabetes	51 (24.6)
Coronary artery disease	51 (24.6)
Previously treated endocarditis	16 (7.7)
Intravenous drug use	10 (4.8)
Preoperative diagnosis (*n* (%))	
Definitive IE	134 (64.7)
Possible IE	73 (35.3)
Echocardiographic findings (*n* (%))	
Vegetations	178 (86.0)
Abscess	33 (15.9)
New valvular regurgitation	154 (74.4)
Culture data	
Positive preoperative blood cultures (*n* (%))	141 (68.1)
Antibiotic treatment	
Treatment > 5 days preoperatively (*n* (%))	105 (52.2)
Days of antibiotic treatment at surgery (mean (sd))	9.7 (12.9)
Pathology (*n* (%))	
Native valve IE	133 (64.3)
Aortic valve	76 (57.1)
Mitral valve	76 (57.1)
Tricuspid valve	11 (8.3)
Prosthetic valve IE	74 (35.7)
Aortic valve	53 (71.6)
Mitral valve	31 (41.9)
Procedure (*n* (%))	
Valve replacement	203 (98)
Aortic valve replacement	130 (62.8)
Mitral valve replacement	98 (47.3)
Tricuspid valve replacement	9 (4.3)
Valve reconstruction	15 (7.2)
Aortic valve repair	2 (0.9)
Mitral valve repair	8 (3.8)
Tricuspid valve repair	6 (2.8)
Clinical outcomes (*n* (%))	
30-day mortality	28 (13.5)

Note: In some cases, multiple heart valves were affected by IE, and combined procedures were performed; the percentages given therefore exceed 100% in total. Abbreviations: BMI: body mass index; COPD: chronic obstructive pulmonary disease, IE: infective endocarditis.

**Table 2 microorganisms-12-01889-t002:** Rates of specific pathogen detection in conventional valve culture and molecular diagnostics (FISHseq) based on preoperative diagnosis, blood culture findings and antibiotic pretreatment.

		Valve Culture		FISHseq		*p*-Value
		Positive	Negative	Positive ^1^	Negative	
All cases (*n* = 207)		68 (32.8)	139 (67.2)	143 (68.1)	64 (31.4)	<0.001
Preoperative diagnosis						
	Definitive IE (*n* = 134)	53 (39.6)	81 (60.4)	103 (76.9)	31 (23.1)	<0.001
	Possible IE (*n* = 73)	15 (20.5)	58 (79.5)	40 (54.8)	33 (45.2)	<0.001
Preoperative blood culture						
	Positive (*n* = 141)	54 (38.3)	87 (61.7)	107 (75.9)	34 (24.1)	<0.001
	Negative (*n* = 52)	7 (13.5)	45 (86.5)	24 (46.2)	28 (53.8)	<0.001
	Inconclusive or missing (*n* = 14)	8 (57.1)	6 (42.9)	12 (85.7)	2 (14.3)	0.094
Type of heart valve						
	Native valve IE (*n* = 133)	43 (32.3)	90 (67.7)	94 (70.7)	39 (29.3)	<0.001
	Prosthetic valve IE (*n* = 74)	25 (33.8)	49 (66.2)	49 (66.2)	25 (33.8)	<0.001
Preoperative antibiotic treatment ^2^						
	≤5 days (*n* = 96)	39 (40.6)	57 (59.4)	71 (74.0)	25 (26.0)	<0.001
	>5 days (*n* = 105)	28 (26.7)	77 (73.3)	67 (63.8)	38 (36.2)	<0.001

Note: All values given as *n* (%). Given *p*-values are based on Pearson’s chi-squared test. ^1^ For this analysis, only cases in which a specific pathogen (species, genus or family) could be detected were considered positive. The positive FISHseq cases include 3 constellations with differing findings in FISH and PCR. ^2^ Data on duration of preoperative antibiotic treatment were missing in 6 cases, resulting in a cohort of *n* = 201 for this analysis.

**Table 3 microorganisms-12-01889-t003:** Results of analyses of the intraoperative specimens by valve culture and molecular biological diagnostics compared to preoperative blood culture findings.

	Valve Culture	FISHseq	
		FISH	PCR
Positive blood cultures (*n* = 141)			
Identical organism	40 (28.4)	58 (41.1)	93 (66.0)
Different organism(s)	12 (8.5)	0	13 (9.2)
Additional organism(s)	1 (0.7)	1 (0.7)	1 (0.7)
No organism	88 (62.4)	82 (58.2)	34 (24.1)
Negative blood cultures (*n* = 52)			
New organism(s)	7 (13.5)	10 (19.2)	22 (42.3)
No organism	45 (86.5)	42 (80.8)	30 (57.7)
Inconclusive blood cultures (*n* = 8)			
Identical organism	0	2 (25.0)	0
Organism partially confirmed	2 (25.0)	2 (25.0)	6 (75.0)
Organism partially confirmed plus an additional organism	2 (25.0)	1 (12.5)	0
Different organism(s)	1 (12.5)	0	1 (12.5)
No organism	3 (37.5)	3 (37.5)	1 (12.5)

Note: All values given as *n* (%). Data on preoperative blood cultures were missing in 6 cases, resulting in a cohort of *n* = 201 for this analysis.

## Data Availability

The data presented in this study are available on request from the corresponding author due to the data protection restrictions accompanying clinical trials.

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
