# Peer review of "Diagnostic Benefit of Molecular Imaging in Patients Undergoing Heart Valve Surgery for Infective Endocarditis"

_microorganisms, 2024, doi:10.3390/microorganisms12091889_

Round 1

Reviewer 1 Report

Comments and Suggestions for Authors

This is a well-perfomed analysis with clear conclusions. Can the FISH method also be used for fungal endocarditis or mixed bacterial-fungal IE? Could account (in part) for the negative outcome in this setting?

Are there indications that application of FISH results in better outcome because of more adequate treatment of IE? 

Reviewer 2 Report

Comments and Suggestions for Authors

Dustin Greve et al. present their research entitled “Diagnostic benefit of molecular imaging in patients undergoing heart valve surgery for infective endocarditis.” The authors conduct a retrospective study aiming to evaluate the potential of FISHseq as an additional molecular biological technique for diagnosing infective endocarditis, compared to conventional valve and blood cultures for pathogen detection. The results show that FISHseq demonstrates higher specificity and sensitivity for detecting certain pathogens. Below, I have some minor concerns regarding this manuscript.

1. Figure 2: How does this black-and-white image indicate bacterial biofilm? It should be using the fluorescence-labeled specific FISH probe to detect the bacterial biofilm, right? I have a similar question regarding Figure 4.

2. Table 2: Please check the sample size for the “Preoperative antibiotic treatment” group. Could you explain why the total number of patients in this category is less than 207?

3. Figure 5: The pie chart indicates a sample size of n = 208. Please verify this.

4. The limitations section should include two additional points:

1) Surgery must be a prerequisite for the FISHseq assay.

2) FISHseq cannot provide information about antibiotic resistance.

5. Please revise the format of the References section to comply with the citation requirements outlined in the “Instructions for Authors.”

Comments on the Quality of English Language

Please pay attention to grammar and improve the clarity of certain sentences.
